

# Symptomatic menopausal transition and risk of subsequent stroke

Chao-Hung Yu[1,*], Chew-Teng Kor[2,*], Shuo-Chun Weng[3,4],
Chia-Chu Chang[5], Ching-Pei Chen[1] and Chia-Lin Wu[2,4,5]

[1] Division of Cardiology, Department of Internal Medicine, Changhua Christian Hospital, Changhua, Taiwan
[2] Internal Medicine Research Center, Changhua Christian Hospital, Changhua, Taiwan
[3] Center for Geriatrics and Gerontology, Division of Nephrology, Department of Internal Medicine, Taichung Veterans General Hospital, Taichung, Taiwan
[4] Institute of Clinical Medicine, National Yang-Ming University, Taipei, Taiwan
[5] Division of Nephrology, Department of Internal Medicine, Changhua Christian Hospital, Changhua, Taiwan
* These authors contributed equally to this work.

Corresponding authors
Ching-Pei Chen,
chingpei@cch.org.tw
Chia-Lin Wu, 143843@cch.org.tw

## ABSTRACT

**Objective:** To examine the long-term risk of stroke in women who have experienced symptomatic menopausal transition.

**Methods:** In this nationwide, population-based cohort study conducted from January 1, 2000 to December 31, 2013, we identified 22,058 women with no prior history of stroke, who experienced symptomatic menopausal transition at ≥45 years of age. Moreover, 22,058 women without symptomatic menopause were matched by propensity scores and enrolled as a comparison group. The propensity score was calculated by using all characteristic variables of each subject, including demographics (age and monthly income), comorbidities (hypertension, hyperlipidemia, diabetes mellitus, obesity, chronic kidney disease, coronary artery disease, congestive heart failure, chronic obstructive pulmonary disease, dysrhythmia, peripheral artery occlusive disease), Charlson's comorbidity index score, clinic visit frequency, and long-term medications (antihypertensives, antidiabetic agents, statins, antiplatelets, aspirin, warfarin, and hormone replacement therapy). The primary endpoint was the development of stroke after the onset of symptomatic menopausal transition. The Fine and Gray's proportional subhazards model was performed to assess the association between symptomatic menopausal transition and subsequent stroke. All subjects were followed up until December 31, 2013.

**Results:** During a mean follow-up of 8.5 years (standard deviation 4.7 years, maximum 14 years), 2,274 (10.31%) women with symptomatic menopausal transition, and 1,184 (5.37%) matched comparison participants developed stroke. The incidence rates were 11.17 per 1,000 person-years in the symptomatic menopausal transition group compared with 8.57 per 1,000 person-years in the comparison group. The risk of developing stroke was significantly higher in women with symptomatic menopausal transition (crude subhazard ratio, 1.31; 95% confidence interval (CI) [1.22–1.41]; $P < 0.001$). After adjusting for demographics, comorbidities, clinic visit frequency, and long-term medications, the risk of stroke remained statistically significant (adjusted subhazard ratio, 1.30; 95% CI [1.21–1.40]; $P < 0.001$). Moreover, subgroup analyses revealed no evidence for inconsistent effects for symptomatic menopausal transition on subsequent risk of stroke across all

subgroups except age, comorbidities, hypertension, and use of antihypertensives. Women with early menopausal transition (before age 50), without comorbid condition, without hypertension, or without use of antihypertensives are at a higher risk of stroke. The longer duration of symptomatic menopausal transition was associated with higher risk of stroke ($P$ for trend < 0.001).

**Conclusion:** In this large-scale retrospective cohort study, symptomatic menopausal transition was statistically significantly associated with a 30% increased risk of stroke. Further prospective studies are required to confirm our findings.

## INTRODUCTION

Menopausal transition is defined as the period before the final menstrual period in late reproductive life. Menopause occurs following 12 months of amenorrhea and represents the loss of ovarian follicular function; this typically occurs in women between 45 and 55 years of age. Approximately 1.5 million women experience menopausal transition every year in the United States (*U.S. Census Bureau, 2005*). Vasomotor symptoms, insomnia, depression, and vaginal dryness are major problems during menopausal transition (*Santoro, Epperson & Mathews, 2015*).

Stroke is the second leading cause of death and third leading cause of disability-adjusted life-years lost worldwide. After stroke, the survivors still have high rates of mortality, recurrent stroke, and disability for months to years (*Hankey, 2017*). The Chinese populations in China and Taiwan were reported to have higher stroke incidence (age-standardized annual first-ever stroke incidence: range 205–584 per 100,000 patients) than white populations (range 170–335 per 100,000 patients) (*Tsai, Thomas & Sudlow, 2013*). Thus, stroke prevention for populations at high risk for stroke remains an important issue in Taiwan.

Symptoms of menopausal transition, such as vasomotor symptoms and night sweats, may have a negative impact on a woman's quality of life during her midlife (*Thurston et al., 2012*). The occurrence of vasomotor symptoms is associated with low serum estrogen levels (*Erlik, Meldrum & Judd, 1982*). In addition, vasomotor symptoms have been found to be associated with chronic insomnia (*Ohayon, 2006*), cardiovascular disease (*Rossouw et al., 2007*), and insulin resistance (*Huang et al., 2017*). A growing body of evidence has also shown the association of vasomotor symptoms with poor control of dyslipidemia in obese women (*Thurston et al., 2012*), atherosclerosis, vascular endothelial dysfunction (*Thurston et al., 2017*), increased prothrombotic events, smoking habits (*Gallicchio et al., 2014*), and disturbed sympathovagal tone (*Thurston et al., 2012*; *Gallicchio et al., 2014*). These deleterious effects accompanied by the symptomatic menopausal transition may potentially contribute to stroke as well as cardiovascular disease. However, no previous studies have assessed the direct link between symptomatic menopausal transition and the risk of stroke with a long-term follow-up. We hypothesized that symptomatic menopausal transition may have insidious adverse effects on the cerebrovascular system.

Therefore, we conducted this nationwide, large-scale, population-based cohort study to determine whether symptomatic menopausal transition is independently linked to stroke.

## MATERIALS AND METHODS

### Data source

Data were retrieved from the National Health Insurance Research Database in Taiwan (NHIRD); this was implemented in 1995 and covers >99% of the population (approximately 23 million people). A database containing one million patients randomly selected from the NHIRD in 2005 and longitudinally linked with NHIRD from 1996 to 2013 was anonymously accessed for research. The comprehensive healthcare information maintained in the database included the date of birth, sex, area of residence, income, ambulatory care, inpatient services, prescription drugs, medical procedures, and diagnostic codes. The International Classification of Diseases, Ninth Revision, Clinical Modification (ICD-9-CM) codes, which have been shown to have high accuracy and validity, were used to identify the diseases (*Cheng et al., 2011*, *2014*; *Yu et al., 2012*). This study was exempt from a full ethical review and was approved by the institutional review board (IRB) of the Changhua Christian Hospital (approval number 190522). The requirement for consent was waived by the IRB.

### Study population

We used a 4-year look-back period (1996–1999) to identify subjects with newly diagnosed symptomatic menopausal transition by excluding pre-existing diagnosis (Fig. 1). Symptomatic menopausal transition was defined by at least three records made by gynecologists within a 1-year period (ICD-9-CM code 627.2: symptomatic menopausal transition). Subjects who were diagnosed with symptomatic menopausal transition during the look-back period were excluded from this study. We identified 457,996 subjects, including 41,516 with newly diagnosed symptomatic menopausal transition and 416,480 without symptomatic menopause, from the NHIRD between January 1, 1996 and December 31, 2013. The index date of the symptomatic menopausal transition group was defined as the first date of the symptomatic menopausal transition diagnosis; this date was also assigned to the matched comparison participants as the date of entry into the study. The exclusion criteria for the study and comparison groups were as follows: (1) age, <45 or >100 years, (2) history of breast cancer before index date (breast cancer treatments can cause menopausal symptoms or premature menopause) (*Cusack et al., 2013*), (3) history of oophorectomy before index date, (4) history of stroke before index date, (5) patients who did not survive or were followed up <30 days (a short exposure time to symptomatic menopausal transition may not be sufficient to cause an incident stroke), and (6) patients who were not matched to the comparison group. Finally, 22,058 women with newly diagnosed symptomatic menopausal transition with no history of stroke were enrolled between January 1, 2000 and December 31, 2013. Additionally, 22,058 comparison participants were selected by propensity score matching. Both groups were followed up until the date of death, developing stroke, withdrawal from the National

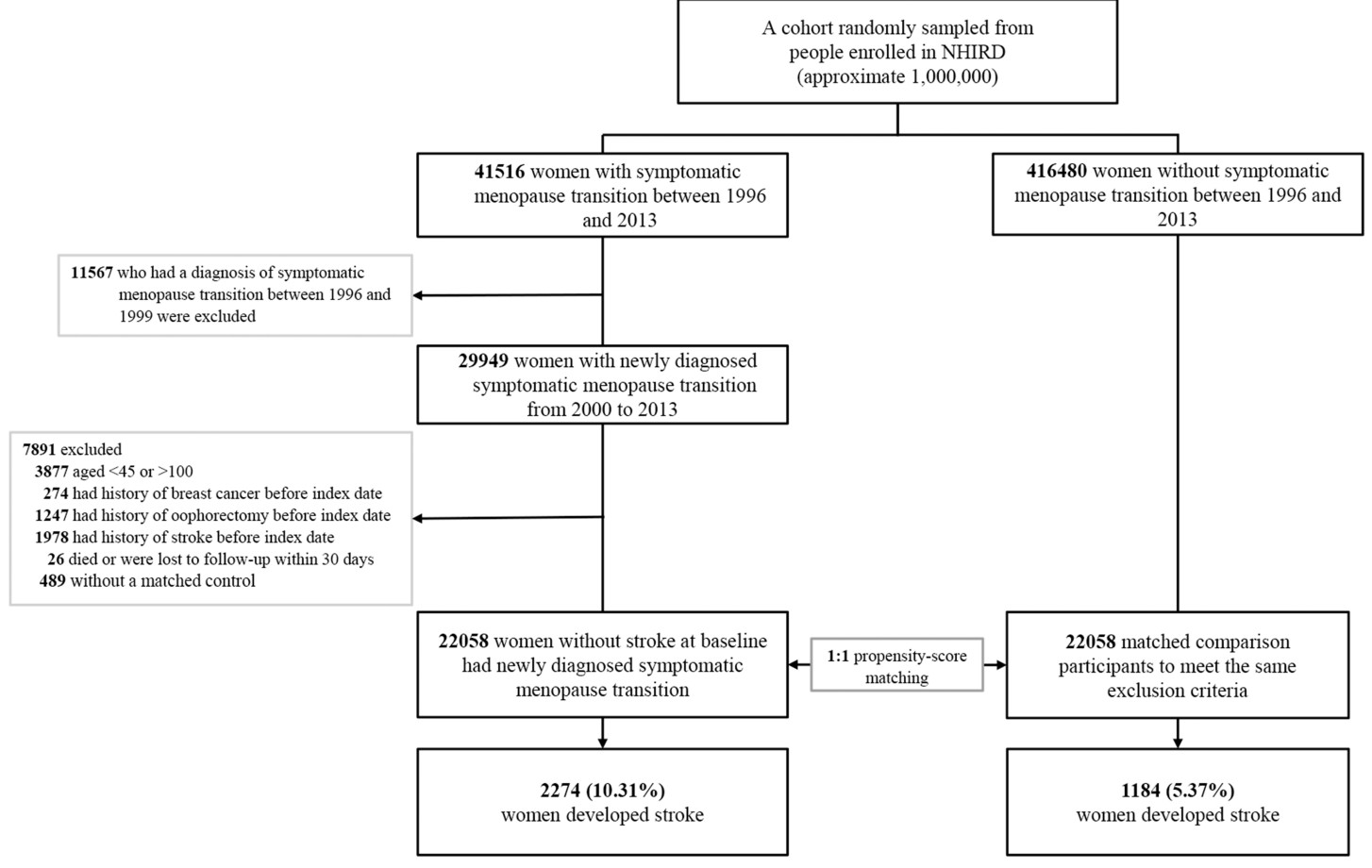

**Figure 1 Study flowchart illustrating the patient selection process and primary outcome.** After exclusion of noneligible subjects, 22,058 women with newly diagnosed symptomatic menopausal transition with no history of stroke were enrolled between January 1, 2000 and December 31, 2013. Additionally, 22,058 matched comparison participants were selected by propensity score matching at a 1:1 ratio. Furthermore, 1,184 women (5.37%) of the comparison group and 2,274 (10.31%) of the symptomatic menopausal transition group developed stroke during the follow-up period. The propensity score was calculated by using all characteristic variables of each subject, including age, monthly income, hypertension, hyperlipidemia, diabetes mellitus, obesity, chronic kidney disease, coronary artery disease, congestive heart failure, chronic obstructive pulmonary disease, dysrhythmia, peripheral artery occlusive disease, Charlson's comorbidity index score, clinic visit frequency, antihypertensives, antidiabetic agents, statins, antiplatelets, aspirin, warfarin, and hormone replacement therapy. NHIRD, National Health Insurance Research Database in Taiwan.

Health Insurance, or the end of 2013. The mean follow-up period was 8.5 years (standard deviation 4.7 years).

## Outcome measures and relevant variables

The primary outcome was the development of stroke following the index date. The stroke (ICD-9-CM codes: 430–438) and comorbidities were diagnosed by at least three records. These diagnostic codes were either from outpatient claims or principal diagnoses of hospitalization records. Potential confounders were included based on previous literature and the possible relationships between them and stroke were examined; these included hypertension, hyperlipidemia, diabetes mellitus, obesity, chronic kidney disease, coronary artery disease, congestive heart failure, chronic obstructive pulmonary disease,

dysrhythmia, peripheral artery occlusive disease, and long-term medications (*Tsai et al., 2016*; *Chen et al., 2015*). We also included additional covariates such as hormone replacement therapy (HRT) and duration of symptomatic menopause. The proxy measures of symptomatic menopause duration were based on the length of time period between the first and last symptomatic menopause-related visits reported by gynecologists.

## Statistical analysis

Propensity score matching was performed to balance the distributions of measured covariates (all variables in Table 1) in the symptomatic menopausal transition and comparison groups. The propensity score was calculated using non-parsimonious multivariable logistic regression, and all variables in Table 1 (age, monthly income, hypertension, hyperlipidemia, diabetes mellitus, obesity, chronic kidney disease, coronary artery disease, congestive heart failure, chronic obstructive pulmonary disease, dysrhythmia, peripheral artery occlusive disease, Charlson's comorbidity index score, clinic visit frequency, antihypertensives, antidiabetic agents, statins, antiplatelets, aspirin, warfarin, and hormone replacement therapy) were included for each patient. Comparison participants were matched on propensity score (nearest-neighbor algorithm with a caliper of 0.1 standard deviation) at a 1:1 ratio (*Austin, 2008*; *Kuo et al., 2015*; *Weng et al., 2017*). The standardized difference (StD) was used to measure the imbalance of all clinical characteristics between the two groups. A StD ≥ 0.1 indicated a significant imbalance between the two groups (*Austin, 2008*). The cumulative incidence of stroke over time was estimated using the cumulative incidence function curves. Deaths prior to the development of stroke were considered as competing risks. Therefore, we conducted the competing risks survival analysis using the Fine and Gray subdistribution hazards model. The association of symptomatic menopausal transition with subsequent development of stroke was reported as subhazard ratios (SHRs) and 95% confidence intervals (CIs). Statistical analyses were conducted using SAS 9.4 software (SAS Institute Inc., Cary, NC, USA). A two-sided *P*-value < 0.05 was considered statistically significant.

## RESULTS

Table 1 shows the baseline characteristics of the two groups. A total of 44,116 women (22,058 women diagnosed with symptomatic menopausal transition and 22,058 matched comparison participants) with no history of stroke were enrolled in the study. The mean age of the enrolled subjects was 52.6 ± 7.4 years. The age, monthly income, comorbidities, Charlson's comorbidity index score, clinic visit frequency, and long-term medications were balanced between the two groups, except a predilection for HRT in the symptomatic menopausal transition group.

### Risk of stroke after perimenopausal transition

During the follow-up period, 1,184 women (5.37%) of the comparison group and 2,274 (10.31%) of the symptomatic menopausal transition group developed stroke (Fig. 1). The incident rates of stroke were 8.57 (95% CI [8.08–9.06] per 1,000 person-years) and 11.17 (95% CI [10.71–11.63] per 1,000 person-years) in the comparison and symptomatic menopausal transition groups, respectively (Table 2). The cumulative incidence of stroke

**Table 1 Demographics and clinical characteristics at baseline.**

| Group | Total (n = 44,116) | Symptomatic menopausal transition (n = 22,058) | Comparison (n = 22,058) | StD[a] |
|---|---|---|---|---|
| Age, mean ± SD, years | 52.6 ± 7.4 | 52.7 ± 6.9 | 52.5 ± 7.8 | 0.03 |
| Monthly income, NTD, n (%) | | | | |
| <15,840 | 19,533 (44.28) | 9,855 (44.7) | 9,678 (43.88) | 0.02 |
| 15,840–25,000 | 15,872 (35.98) | 7,717 (35.0) | 8,155 (36.97) | 0.04 |
| 25,000 | 8,711 (19.75) | 4,486 (20.3) | 4,225 (19.15) | 0.03 |
| Charlson's comorbidity index score | | | | |
| Mean ± SD | 0.88 ± 1.24 | 0.89 ± 1.19 | 0.88 ± 1.29 | 0.01 |
| Median (IQR) | 0 (0–1) | 0 (0–1) | 0 (0–1) | |
| 0 (n (%)) | 22,937 (52.0) | 11,084 (50.3) | 11,853 (53.7) | 0.07 |
| 1–2 (n (%)) | 16,847 (38.2) | 8,891 (40.3) | 7,956 (36.1) | 0.09 |
| ≥3 (n (%)) | 4,332 (9.8) | 2,083 (9.4) | 2,249 (10.2) | 0.03 |
| Clinic visit frequency, visits per year | | | | |
| Mean ± SD | 25.65 ± 18.03 | 25.87 ± 16.72 | 25.44 ± 19.25 | 0.02 |
| Median (IQR) | 22 (13–34) | 23 (14–34) | 21 (11–35) | |
| Comorbidity, n (%) | | | | |
| Hypertension | 8,258 (18.7) | 4,171 (18.9) | 4,087 (18.5) | 0.01 |
| Hyperlipidemia | 4,801 (10.9) | 2,443 (11.1) | 2,358 (10.7) | 0.01 |
| Diabetes mellitus | 2,979 (6.8) | 1,508 (6.8) | 1,471 (6.7) | 0.007 |
| Obesity | 244 (0.6) | 111 (0.5) | 133 (0.6) | 0.01 |
| CAD | 2,677 (6.1) | 1,383 (6.3) | 1,294 (5.9) | 0.02 |
| CHF | 515 (1.2) | 261 (1.2) | 254 (1.2) | 0.003 |
| Dysarrhythmia | 1,458 (3.3) | 744 (3.4) | 714 (3.2) | 0.008 |
| COPD | 3,552 (8.1) | 1,778 (8.1) | 1,774 (8.0) | <0.001 |
| CKD | 884 (2) | 449 (2.0) | 435 (2.0) | 0.005 |
| PAOD | 213 (0.5) | 102 (0.5) | 111 (0.5) | 0.006 |
| Long-term use medications[b], n (%) | | | | |
| Antihypertensives | 5,353 (12.1) | 2,695 (12.2) | 2,658 (12.1) | 0.005 |
| Antidiabetic agents | 1,710 (3.9) | 863 (3.9) | 847 (3.8) | 0.004 |
| Statins | 1,467 (3.3) | 753 (3.4) | 714 (3.2) | 0.01 |
| Hormone replacement therapy | 5,383 (12.2) | 4,916 (22.3) | 467 (2.1) | 0.65 |
| Estrogens | 2,863 (6.5) | 2,591 (11.8) | 272 (1.2) | 0.44 |
| Progestogens | 2,051 (4.7) | 1,845 (8.4) | 206 (0.9) | 0.36 |
| Combined progestogens and estrogens | 2,362 (5.4) | 2,174 (9.9) | 188 (0.9) | 0.41 |
| Antiplatelets | 1,607 (3.6) | 818 (3.7) | 789 (3.6) | 0.007 |
| Aspirin | 1,169 (2.7) | 561 (2.5) | 608 (2.8) | 0.01 |
| Warfarin | 75 (0.2) | 36 (0.2) | 39 (0.2) | 0.003 |
| Propensity score (mean ± SD) | 0.3 ± 0.14 | 0.3 ± 0.14 | 0.3 ± 0.14 | <0.001 |

Notes:
CAD, coronary artery disease; CHF, congestive heart failure; CKD, chronic kidney disease; COPD, chronic obstructive pulmonary disease; IQR, interquartile range; NTD, new Taiwan dollars; PAOD, peripheral artery occlusive disease; SD, standard deviation; StD, standardized difference.
[a] Standardized difference (StD) of greater than 0.1 is considered important imbalance.
[b] Defined as drug prescription for at least 3 consecutive months.

**Table 2 Incidence and risk of stroke in patients with symptomatic menopausal transition and their matched subjects.**

| Group | Event | PY | Incidence[a] | Model 1 | | Model 2 | | Model 3 | |
|---|---|---|---|---|---|---|---|---|---|
| | | | | cSHR[b] (95% CI) | P-value | aSHR[b] (95% CI) | P-value | aSHR[b] (95% CI) | P-value |
| Comparison | 1,184 | 138,138.72 | 8.57 (8.08–9.06) | Reference | | Reference | | Reference | |
| Symptomatic menopausal transition | 2,274 | 203,586.37 | 11.17 (10.71–11.63) | 1.31 [1.22–1.41] | <0.001 | 1.37 [1.27–1.48] | <0.001 | 1.30 [1.21–1.40] | <0.001 |

Notes:
Model 1: crude hazard ratio compared with the propensity-score matched comparison subjects.
Model 2: adjusted for all variables listed in Table 1.
Model 3: adjusted for all variables listed in Table 1, as well as comorbidities and medications considered as time-dependent covariates.
aSHR, adjusted subhazard ratio; cSHR, crude subhazard ratio; CI, confidence interval; PY, person-years.
[a] Per 1,000 person-years.
[b] Death before developing stroke was considered a competing risk.

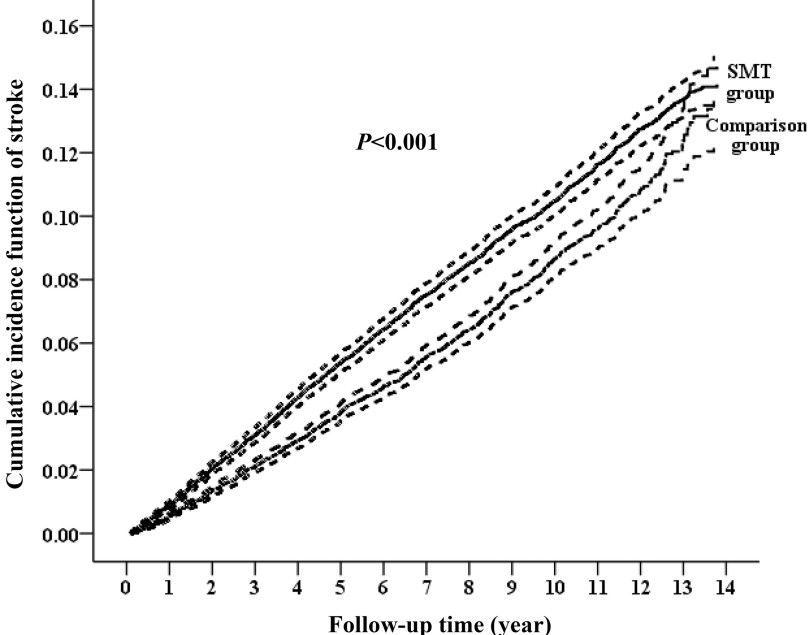

**Figure 2 Cumulative incidence function curves with 95% confidence intervals for the risk of subsequent stroke between the two groups.** The cumulative incidence of stroke was significantly higher in the symptomatic menopausal transition group (SMT) than in the comparison group.

was higher in the symptomatic menopausal transition group than in the comparison group (Fig. 2). The crude risk of stroke in the symptomatic menopausal transition group was statistically significantly higher compared to the comparison group (SHR, 1.31; 95% CI [1.22–1.41]; $P < 0.001$). Additionally, two multivariable models were used to adjust the risk of subsequent stroke. After adjusting for all confounders in Table 1 (Model 2, Table 2) and the full adjustment model (Model 3, Table 2), the risk of subsequent stroke remained consistently higher in the symptomatic menopausal transition group than that of the comparison group (SHR, 1.37; 95% CI [1.27–1.48]; $P < 0.001$ and SHR, 1.30; 95% CI [1.21–1.40]; $P < 0.001$, respectively).

**Table 3 Risk of stroke in patients with symptomatic menopausal transition and their matched subjects regarding the age of onset, comorbidity, and hormone replacement therapy.**

| Subgroup | Comparison | | Symptomatic menopausal transition | | Symptomatic menopausal transition vs. comparison | | | | |
|---|---|---|---|---|---|---|---|---|---|
| | $n$ | Event | $n$ | Event | Model 1 aSHR (95% CI)[a,b] | $P$-value | Model 2 aSHR (95% CI)[b,c] | $P$-value | $P_{interaction}$[d] |
| Age, years | | | | | | | | | <0.001 |
| <50 | 10,534 | 237 | 8,712 | 490 | 1.55 [1.31–1.83] | <0.001 | 1.42 [1.21–1.67] | <0.001 | |
| 50–64 | 9,403 | 527 | 11,647 | 1,289 | 1.30 [1.16–1.45] | <0.001 | 1.29 [1.16–1.43] | <0.001 | |
| ≥65 | 2,121 | 420 | 1,699 | 495 | 1.20 [1.03–1.39] | 0.02 | 1.19 [1.03–1.37] | 0.02 | |
| Comorbidity | | | | | | | | | <0.001 |
| 0 | 15,409 | 515 | 14,348 | 984 | 1.58 [1.41–1.78] | <0.001 | 1.41 [1.26–1.58] | <0.001 | |
| ≥1 | 6,649 | 669 | 7,710 | 1,290 | 1.19 [1.07–1.32] | 0.001 | 1.14 [1.03–1.26] | 0.01 | |
| Hormone replacement therapy | | | | | | | | | 0.66 |
| No | 21,591 | 1,146 | 17,142 | 1,637 | 1.35 [1.25–1.47] | <0.001 | 1.27 [1.17–1.38] | <0.001 | |
| Yes | 467 | 38 | 4,916 | 637 | 1.28 [0.92–1.79] | 0.15 | 1.25 [0.90–1.75] | 0.19 | |

Notes:
aSHR, adjusted subhazard ratio; CI, confidence interval.
[a] Adjusted for all variables listed in Table 1.
[b] Death before developing stroke was considered a competing risk.
[c] Adjusted for all variables listed in Table 1, as well as comorbidities and medications considered as time-dependent covariates.
[d] $P$-values for interactions were obtained from Model 2.

## Risk of stroke by age, comorbidities, and HRT

The SHR for subsequent stroke was statistically significantly higher in women with symptomatic menopause compared to comparison participants; this was the case for young (<50 years), middle-aged (50–64 years), and elderly (≥65 years) women. However, the interaction between symptomatic menopausal transition and age category was statistically significant ($P_{interaction}$ < 0.001, Table 3). The risk of developing stroke tended to decrease with increasing age (SHRs from 1.42 to 1.29 to 1.19). Additionally, there was a statistically significant interaction between symptomatic menopausal transition and comorbidity ($P_{interaction}$ < 0.001, Table 3). Symptomatic menopausal women without comorbid condition had a higher risk of developing stroke than those with any comorbidity (SHRs, 1.41 vs. 1.14). However, the interaction between the use of HRT and stroke was not statistically significant ($P_{interaction}$ = 0.66, Table 3) though the SHR for stroke was only statistically significantly higher in symptomatic menopausal women without HRT compared to comparison participants.

## Subgroup analyses

The risk of subsequent stroke was further estimated throughout the baseline clinical characteristics, including individual comorbidities and long-term medications (Fig. 3). We found that there was no statistically significant interaction between symptomatic menopausal transition and all subgroups except hypertension and use of antihypertensives ($P_{interaction}$ < 0.001 and 0.003, respectively). The risk of stroke was further increased in symptomatic menopausal women without hypertension or without the use of antihypertensives (Fig. 3).

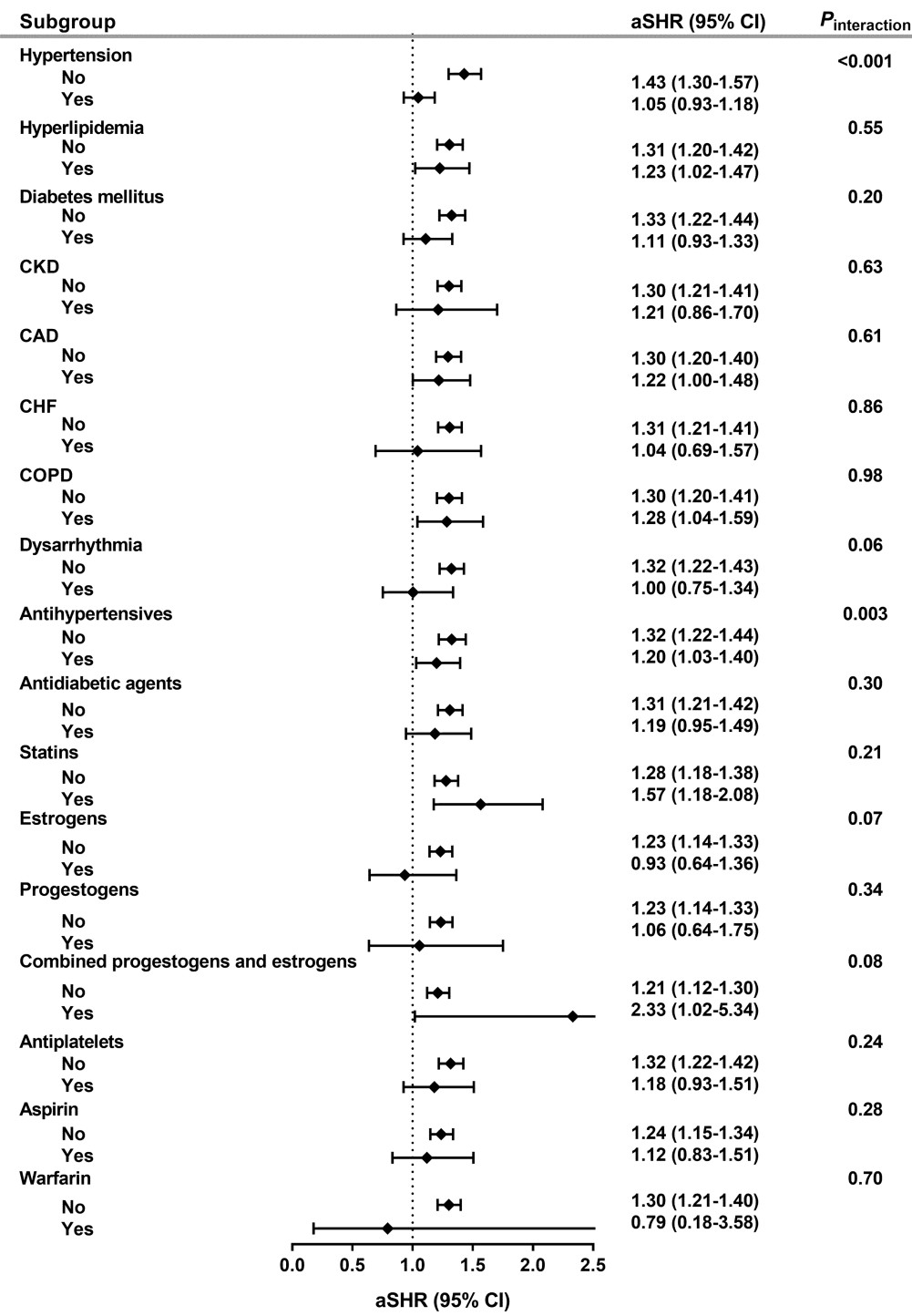

**Figure 3 Subgroup analyses.** The risk of subsequent stroke was consistent across all subgroups except hypertension and use of antihypertensives. aSHR, adjusted subhazard ratio; CAD, coronary artery disease; CHF, congestive heart failure; CI, confidence interval; CKD, chronic kidney disease; COPD, chronic obstructive pulmonary disease.

**Table 4 Risk of stroke in patients with symptomatic menopausal transition compared with the comparison group regarding the duration of symptomatic menopause.**

| Duration of symptomatic menopause (years) | aSHR (95% CI)[a,b] | P-value | aSHR (95% CI)[b,c] | P-value |
|---|---|---|---|---|
| 0 | Reference | – | Reference | – |
| 0–1.4 | 1.20 [1.09–1.33] | <0.001 | 1.18 [1.07–1.29] | 0.001 |
| 1.4–4.8 | 1.29 [1.17–1.43] | <0.001 | 1.26 [1.14–1.39] | <0.001 |
| >4.8 | 1.41 [1.28–1.55] | <0.001 | 1.33 [1.21–1.45] | <0.001 |
| P for trend | | <0.001 | | <0.001 |

Notes:
aSHR, adjusted subhazard ratio; CI, confidence interval.
[a] Adjusted for all variables listed in Table 1.
[b] Death before developing stroke was considered a competing risk.
[c] Adjusted for all variables listed in Table 1, as well as comorbidities and medications considered as time-dependent covariates.

## Duration of symptomatic menopausal transition and stroke

We estimated the association of the duration of symptomatic menopausal transition with the development of subsequent stroke. The longer duration of symptomatic menopausal transition was associated with higher risk of stroke (Table 4).

## DISCUSSION

The present study is the first and largest study to investigate the association between symptomatic menopausal transition and subsequent development of stroke during up to 14 years of follow-up. In this prognostic model, symptomatic menopausal transition has added statistically significantly prognostic power for stroke to standard risk factors. Women with symptomatic menopausal transition had a 30% increased risk of subsequent stroke compared to matched comparison participants. The effect of symptomatic menopause on stroke was consistent across all subgroups except age, comorbidity, hypertension, and the use of antihypertensives. Moreover, this effect was durable and increased when the duration of menopausal transition lasted longer.

Previous studies have shown that there is a possible link between stroke and symptomatic menopausal transition. Women experiencing vasomotor symptoms tend to have lower circulating estradiol levels and other adverse cardiovascular risk factors such as obesity, smoking, and psychosocial problems (*Gast et al., 2010*). Among them, low socioeconomic status, depression, and anxiety have the closest relationship between these factors and cardiovascular diseases (*Thurston et al., 2012*). Additionally, dysregulation of adipocyte-derived hormones, especially the higher circulating leptin levels, lower adiponectin levels, and higher leptin-to-adiponectin ratios, was reported in women transitioning through menopause (*Ben Ali et al., 2011*). A recent study also showed that the profile of adipokines (leptin levels, adiponectin levels, and leptin-to-adiponectin ratios) was closely related to the severity of vasomotor symptoms (*Huang et al., 2017*). Leptin, for example, is a widely investigated adipokine and its production is proportional to the adipose tissue mass and its secretion is pulsatile with the characteristics of diurnal rhythm. The serum leptin level is highest at night and lowest in the morning; as the leptin level rises, it increases the appetite via its effects on the brain, resulting in weight gain and obesity

(*Kelesidis et al., 2010*). Moreover, both menopause and low levels of estrogens can cause systemic inflammation and neuroinflammation (*Au et al., 2016*). Therefore, a low estrogen level, dysregulation of adipokine production, high BMI, and proinflammatory status may be potential underlying pathogenic factors for stroke in women with symptomatic menopausal transition.

In the present study, the risk of stroke was especially high in women with early onset of symptomatic menopause, not using antihypertensives, in the absence of comorbidity, and without hypertension. *Lisabeth et al. (2009)* reported that natural menopause before age 42 was associated with a higher risk of ischemic stroke. Another study also suggested that early menopause is associated with an increased risk of ischemic stroke (*Rocca et al., 2012*). Early menopause may expose women to longer duration of systemic- or neuro-inflammation contributing to a higher risk of subsequent stroke. We also observed an interaction between symptomatic menopausal transition and comorbidity whereby the risk of stroke was highest in women without any comorbidity and the effect of symptomatic menopausal transition on stroke was significantly reduced in women with a comorbid condition. Furthermore, since blood pressure control is one of the most important risk factors for stroke, especially ischemic stroke in women, we evaluated the interaction between symptomatic menopausal transition and hypertension or use of antihypertensives (*Gorgui et al., 2014*). Our results revealed that the risk of stroke was much higher in women without or without control of hypertension. Because hypertension is already a strong risk factor for stroke, the impact of symptomatic menopausal transition on subsequent stroke becomes smaller in the hypertension or with antihypertensives subgroups than in the non-hypertension or without antihypertensives subgroups. Our findings suggest that these subgroups are at the highest risk for stroke and that the prevention of subsequent stroke should be considered in addition to traditional management of symptomatic menopause.

Vasomotor symptoms are some of the major problems leading middle-aged women to seek medical advice and treatment for symptomatic menopause. Although HRT plays a key role in treating symptomatic menopausal transition (*Erlik, Meldrum & Judd, 1982*; *Huang et al., 2008*), HRT has been reported to increase the risk of breast cancer (*Russo & Russo, 2006*), cerebral (*Rossouw et al., 2002*) or cardiac vascular events (*Wilson, Garrison & Castelli, 1985*), and new-onset atrial fibrillation (*Tsai et al., 2016*; *Bretler et al., 2012*). We found that the risk of stroke remained higher in women with symptomatic menopausal transition after adjustment for HRT. Furthermore, there was no statistically significant interaction between symptomatic menopausal transition and HRT.

The strength of our study is its large sample size. A large sample size is more likely to produce sufficiently narrow CIs. In this case, the narrow CIs of our results (e.g., 95% CI of SHR [1.21–1.40]; Table 2) indicate adequate precision. We did not conduct ex ante power calculations because our study utilized medical claims retrospectively retrieved from the NHIRD. However, the retrospective power of this study calculated using an observed effect size was determined entirely by the *P*-value which was already observed. Thus, post hoc sample size calculation or power analysis provides no additional information beyond the effect size. Additionally, the number of events observed may be

important when modeling survival data. Previous studies have shown that 10–50 events per variable may be necessary to assure accurate estimation of regression coefficients, standard errors, and CIs (*Austin, Allignol & Fine, 2017*; *Peduzzi et al., 1995*). A 14-year follow-up was more than appropriate to detect stroke events in middle-aged women (3,458 stroke events were observed during the study period). However, there were some limitations to this study. First, the NHIRD did not include information relevant to stroke such as smoking history, BMI, physical activity, daily salt intake, blood pressure, and blood glucose and lipid profiles. These unmeasured covariates could have an impact on the outcome, even after balancing the baseline clinical characteristics with propensity score matching. Second, the diagnosis of symptomatic menopausal transition was mainly based on the ICD-9-CM codes; this could lead to the underestimation of cases with less severe symptoms. However, the risk of stroke may be underestimated if the cases were misclassified as matched comparison participants. Third, we cannot exclude the "healthy user bias"; patients who seek medical advice might be compliant with healthy lifestyles or treatments that could reduce the occurrence of adverse events (*Brookhart et al., 2007*). Finally, since most of the population in Taiwan are Han Chinese, further studies are required to clarify whether these findings can be generalized to other ethnicities.

## CONCLUSION

This large-scale long-term cohort study demonstrated that symptomatic menopausal transition was statistically significantly associated with an increased risk of subsequent stroke. Women with early menopausal transition, without comorbid condition, without hypertension, or using antihypertensives are at a higher risk of stroke. In order to fully elucidate whether stroke prevention improves cerebrovascular outcomes in the care of symptomatic menopausal transition further studies are required.

### Funding
This study was funded by grants 106-CCH-IRP-022, 107-CCH-NFP-002, and 107-CCH-HCR-031 from the Changhua Christian Hospital Research Foundation. The funders had no role in study design, data collection and analysis, decision to publish, or preparation of the manuscript.

### Grant Disclosures
The following grant information was disclosed by the authors:
Changhua Christian Hospital Research Foundation: 106-CCH-IRP-022, 107-CCH-NFP-002, and 107-CCH-HCR-031.

### Competing Interests
The authors declare that they have no competing interests.

## Author Contributions

- Chao-Hung Yu conceived and designed the experiments, prepared figures and/or tables, authored or reviewed drafts of the paper, approved the final draft.
- Chew-Teng Kor conceived and designed the experiments, performed the experiments, analyzed the data, contributed reagents/materials/analysis tools, prepared figures and/or tables, approved the final draft.
- Shuo-Chun Weng conceived and designed the experiments, authored or reviewed drafts of the paper, approved the final draft.
- Chia-Chu Chang conceived and designed the experiments, contributed reagents/materials/analysis tools, approved the final draft.
- Ching-Pei Chen conceived and designed the experiments, contributed reagents/materials/analysis tools, authored or reviewed drafts of the paper, approved the final draft.
- Chia-Lin Wu conceived and designed the experiments, contributed reagents/materials/analysis tools, prepared figures and/or tables, authored or reviewed drafts of the paper, approved the final draft.

## Human Ethics

The following information was supplied relating to ethical approvals (i.e., approving body and any reference numbers):

The Institutional Review Board of the Changhua Christian Hospital approved this study (approval number 190522).

## Data Availability

The raw data and codebook are available as Supplemental Files.

## Supplemental Information

Supplemental information for this article can be found online at http://dx.doi.org/10.7717/peerj.7964#supplemental-information.

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
