# Peer review of "Symptomatic menopausal transition and risk of subsequent stroke"

_PeerJ, doi:10.7717/peerj.7964_

## Round 0.1 · original submission · Major Revisions

Dear Authors, There are major issues that have to be addressed in the revised manuscript. Thanking you

Reviewer 1 ·

Basic reporting

In this study, Yu and colleagues analyzed a large-scale cohort to uncover the impact of symptomatic menopausal transition on the risk of stroke. They concluded symptomatic menopausal transition could significantly increase the risk of stroke. In general, the experiment was designed properly and this manuscript was well written with sufficient introduction and discussion. I have some suggestions to the authors:

1. Line 87-88: Could the authors specify the reason that they used a 4-year look-back period (1996–1999) in this study and excluded 11567 cases (Figure 1)?

2. Previous studies showed that age at menopause could influence the risk of ischemic stroke. Did this factor play a role in the current study? Could the authors also cite and discuss the work done by Lisabeth et. al., 2009 (Stroke)? Is that possible some cases in “control group” had early menopause?

Experimental design

no comments

Validity of the findings

no comments

Additional comments

no comments

Reviewer 2 ·

Basic reporting

Present study by YU et al examined the symptomatic menopausal occurrence, transition, and following stroke incidence. Fourteen years follow-up large retrospective cohort study compared the development of stroke between symptomatic and non-symptomatic menopausal women. Author found that this symptomatic menopausal transition strongly associated with increased risk of stroke.

1. Largest cohort is plus of this study. Appreciated.
2. On other hand, it is retrospective study.

Experimental design

1. Data collected for 14 years
2. Data acquired from national health insurance research database in Taiwan
3. Study approved by IRB and does not require ethical review.
4. Study examined 22058 control and 22058 symptomatic menopause women.

Validity of the findings

Women with symptomatic menopause had 30% increase in risk of stroke.
Overall, study has given reasonable explanation on statistical basis.

Additional comments

1. How did author adjust demographics with stroke incidence?
2. There would discrepancies menopause women undergoing HRT. Besides, the time of initiation of HRT do matters in terms of vascular reactivity. Why would not author consider excluding HRT from the cohort?
3. Please include figure legends
4. Hypertension is one of the main causes of stroke and it is positively correlated with symptomatic menopause. But author wrote “The risk of stroke was further increased in symptomatic menopausal women without hypertension or without the use of hypertensives” It is not clear what author trying to convey.
5. Line 172 & 208: is it use of hypertensives or anti-hypertensives?
6. Is there any relation in leptin concentration on menopause or stroke?
7. How did author analyzed hypertension if the BP value is not given?
8. Are the current cohort used same individuals as “Effects of early age at natural menopause on coronary heart disease and stroke in Chinese women.”?

·

Basic reporting

The article in general has very high quality. The figures and tables are well made with clear information.

It might be constructive and beneficial if in the introduction part, the authors can discuss in more details in a more compelling way on why they want to “determine whether symptomatic menopausal transition is a contributing factor to stroke” (line 69-70). The authors claim that “[n]o previous studies have assessed the direct link between symptomatic menopausal transition and the risk of stroke with a long-term follow-up,” (line 67-68) and “[t]herefore” they wanted to look at stroke. However, they have never discussed specifically about stroke in the introduction. The authors do have discussed a possible linked between menopausal transition and cerebrovascular problems (line 57-66). But cerebrovascular problems are not limited to stroke and also include transient ischemic attack, aneurysms, and vascular malformations. So why specifically looking at stroke? The way the authors put it as in line 67-70 makes it seems like it is because no one ever did such that they wanted to do so.

The writing in general is clear but does have several places that are ambiguous/misleading, two of which are discussed in the third section of my review (Validity of the findings) as they are related to the conclusions the authors made. Another place is in line 52-53, where the authors write “[d]uring menopausal transition, vasomotor symptoms, insomnia, depression, and vaginal dryness are major problems that affect 1.5 million women every year.” The message from this sentence is that 1.5 million women every year experience the major problems of menopausal transition. However, according to the reference (Santoro et al., 2015), 1.5 million women experience menopausal transition, but may not all suffer from the listed major problems. It would be inaccurate if the authors rephrase the previous study’s finding in the way that they’ve done in line 52-53.

Experimental design

The experiment design is very solid. The sample sizes of both the disease group and the control group are impressive, and the time scope of the follow-up period is remarkable, both of which make the results very credible.

However, it is unclear to me that when the authors look at the relationship between the duration of symptomatic menopausal transition and stroke, why they use the number of visits to gynecology clinics to represent duration of symptomatic menopausal transition, rather than, for example, using the length of time period since the subjects first went to gynecology clinics through the last time they did so on record. The number of visits cannot speak for the duration unless all subjects have the same frequency of visits to gynecology clinics, which is not mentioned in the article.

In addition, it would make the methods they use more transparent if they can put in more details about the propensity score matching procedure that they performed in order to select the controls. It was only mentioned briefly that they use "measured covariates" (line 119-120) for the calculation, but what exact covariates do they use?

Validity of the findings

The findings are statistically sound and controlled. And their discussion on the potential pitfalls of the study is very neat.

However, the conclusion in the article (not the one in the abstract) is not very well stated and is misleading. In line 249-250, the authors claimed that "symptomatic menopausal transition significantly increased the risk of subsequent stroke." The use of the word "increased" indicated that there is a causation relationship between symptomatic menopausal transition and the risk of subsequent stroke. Yet this study is only an association study and there is no proof that symptomatic menopausal transition would cause the risk of stroke to increase. I think the conclusion in line 249-250 should be stated the same way as they have done in the abstract (line 43-44).

A same mistake also appears in line 209-210 when the authors are referring to previous findings and write "early menopause may increase the risk of ischemic stroke." This is again misleading since the verb "increase" would imply causation, and in fact, nowhere in the reference (Rocca et al., 2012) it was directly stated that early menopause would increase ischemic stroke risk but only claimed that the two are associated.

Additional comments

Overall I enjoyed reading this article. If my comments as above are addressed, I recommend the article for publication.

·

Basic reporting

The abstract is very short and I’ll suggest some additional information, and other edits, that it might be useful to consider here:

Line 28: This could potentially be read as the period of entry into the cohort (between 2000 and 2013) or as suggesting a cohort begun in 2000 and followed for up to 14 years. Could you make this clearer, which would be especially helpful for the reader who only reads or focuses on the abstract? Related to this, it would be useful to add the mean years of follow to the abstract’s (below) and the manuscript’s (later) results, I think.

Line 31: I wonder if you could list the variables used for the propensity score matching here. As noted, the abstract is short, so there is plenty of room for additional detail, but perhaps this would be more readable if you collapsed the individual comorbidities and long-term medications into single list items each.

Line 31: Perhaps “…as a comparison group…” (adding “a”, replacing “control” with “comparison”, and using “group” rather than “cohort” which I’d interpret as both groups combined), or simply adding “the” would also work.

Lines 31–32: The primary outcome, assuming this means the outcome variable, would just be stroke as it applies to both groups. If the “primary outcome” here is referring to a research question, the wording would be different again.

Line 33: You give an exact date (month-day-year) for the end of the cohort follow-up, so could perhaps indicate the same back on Line 28 for the start of the cohort period.

Line 33: I think a sentence or two around the statistical methods used would be very helpful for the reader here.

Line 38: I think I know what you mean, but I’m not sure that it “remained” significant when you haven’t presented any results showing significance above. You could add the crude SHR beforehand (to show the statistical significance remaining) or reword this to note that it was significant when adjusting for these variables.

Line 38: Isn’t this a sub-hazard ratio? This applies throughout the manuscript and tables.

Line 40: Perhaps “no evidence for effect modification by…” or “no evidence for inconsistent effects for…” so that this is worded in terms of the lack of evidence rather than a claim of a lack of meaningful numerical difference (which is how I’d interpret “consistent effects”).

Line 41: I think a reader of the abstract would be interested in which age, comorbidity, etc. groups, had higher or lower risk than others.

Line 42: I’m not sure “durable” is the right word here, perhaps “There was no evidence that the effect varied by the duration of symptomatic menopausal transition.”? See also Line 186.

Line 44: I’d qualify “significantly” here as “statistically significantly”, unless you mean “statistically and clinically significantly”. I think it’s worth checking the manuscript for any instances of “significant” (or its inflected forms) where there is even the slightest ambiguity as to whether this is statistical, practical/clinical, or both.

In the background, Line 50, I wonder if it would be possible to clarify here that 12 months without a menstrual period is part of the definition of menopause, similar to how you define menopausal transition on Line 49. Also, for Line 53, you could qualify where these “1.5 million women every year” are from (e.g. “1.5 million women in [country] every year…”). Note that in the reference you provide for this figure, the figure itself appears to be referenced to an earlier article and, if I’m correct in this, I think the original source should be cited.

Lines 98–99: This might be clearer as “patients who did not survive for at least 30 days after their index date or were followed up for fewer than 30 days after this date” if I’m understanding your intention (at the moment “did not survive” is potentially unqualified). What was the reason for this exclusion and could you add this to the manuscript?

On Line 122, when you say “considered” do you mean “included”, rather than being a candidate for inclusion depending on some criterion or criteria?

Can you add a reference for Lines 123–124?

For Lines 159–162, I think I’d show the significance of the interaction first, as the first sentence could be (mis)read as reporting the age main effects until you reach the interaction p-value at the end. The same issue arises later on Lines 169–171.

For Line 160, a test for linear trend (via orthogonal polynomials) would be useful to support the description here (“decreased”) rather than a single (Wald?) p-value which would only indicate “differed”. Or you could discuss the pairwise comparisons here also alongside the interaction p-value.

Lines 165–166: I wouldn’t consider this to be a valid interpretation based on a p-value only. There is no evidence that the effect of symptomatic menopausal transition on the risk of subsequent stroke varied by HRT, but there is also no direct evidence for a consistent effect (although you could consider whether the interaction’s 95% CIs are consistent with meaningful effect modification or not). A similar point applies to Line 170 and a related one to Lines 229–230 (the final clause here is fine, it’s the use of “similarly higher” that isn’t supported by the p-value alone).

Why no interaction p-value for duration (Lines 176–178 and Table 4)? Again, a linear trend interaction would be an option (while testing for interactions with and main effects for higher-order polynomials as well).

Lines 182–183: Isn’t this “during up to 14 years of follow-up”?

It might be useful to add the variables used for the propensity scores to Figure 1 (possibly using the suggested shortening for comorbidities and medication suggested above).

I’m not sure how easy SAS makes this, but adding 95% CIs would be useful for Figure 2, and with only two groups this shouldn’t make the figure too noisy.

Table 1 might work better with the columns: total, control, symptomatic menopausal transition, and then StD so that the difference column is directly alongside the two columns that it refers to.

Also in Table 1, note that Charlson’s scores and clinic visits are both positively skewed and I don’t think means and SDs are likely to be appropriate here; perhaps you could show medians and IQRs?

In Tables 2, 3, and 4, as far as I can tell, all of the HRs and in fact SHRs. Is there a reason for not labelling them as this?

A data dictionary and, if possible, the SAS code used for the analyses would be very useful complements to the actual data for the reader trying to replicate the results in the manuscript. For example, I assume that com_stroke==2 indicates non-stroke mortality, but it would be good to have such things made clear.

Experimental design

What was the reason for selecting patients in 2005 (Line 76)? Presumably, patients who died (including following stroke) between 2000 and 2005 were thus rendered ineligible for inclusion in the study cohort, whereas a women who did not die during this period could be included in either group starting as early as 2000. This would seem to bias any excess hazard from symptomatic menopausal transition downwards (survivor bias), or am I missing something here that would prevent this?

When you say (Lines 95–96): “this date was also assigned to the matched control”, I’m assuming you mean for entry into the cohort? If so, could you make this explicit here?

Line 97: I’m not entirely sure why you’d need an age>100 criterion here.

Line 97: Could you explain, also in the manuscript, why you excluded those with a history of breast cancer prior to the index date? This might also lead to edits on Lines 29 and 101 (and perhaps elsewhere) to add history of breast cancer to these sentences.

I think some additional information is needed in the statistical methods. It’s not clear to me why you have used both Cox’s proportional hazards (Lines 125–128) and Fine and Gray’s model (Lines 129–130). I’m assuming participants were censored at death for the former model (which could be made clearer if this is the case) but I can’t see any reported results for the Cox’s models. The three sets of models described on Lines 150–152 (not the results, just the sets of variables in each model) seem like they should have been included here in the methods along with details of the interactions investigated (are the associated p-values for interactions from Wald tests?) Also, did you test for time-varying effects for the group, propensity score, or other variables included? And if so, how? Did you look at any other residual diagnostics for the models?

What was the basis for the sample size chosen? In the discussion, you describe a priori power for some undescribed effect size (Line 233). This should be included in the statistical methods with sufficient information to enable a reader to replicate the calculation. This might include some consideration of maximal model complexity (e.g. based on Peduzzi, et al.’s guideline).

I also suggest listing the variables used on Line 121 as well as referring to Table 1 to save the reader needing to navigate to the table for this; and combining the two method references to the details of the propensity scores (Lines 102–103 and Lines 119–124) into one place so that all of this information is together. For the latter point, I’d move the algorithm and caliper to the second reference, but this is up to you.

I’m not able to follow your reasoning for adjusting for the propensity score and then in another model adjusting for the variables used to create the propensity score (Models 2 and 3 in Table 2, and Lines 151–152, which I’ve suggested moving to the methods). Can you explain?

Validity of the findings

How confident are you that none of the “confounders” listed on Lines 111–114 are on the causal pathway between symptomatic menopausal transition and stroke? If you have drawn a DAG for this, it would be worth considering including this as a supplement. The underlying causal model here seems quite complex!

The statistical methods talk about using both Cox’s proportional hazards regression, which might be valid from an aetiological perspective but not from a prognostic one, and competing risks analysis. However, all effect sizes are labelled as “HR” and it is unclear in the text which results are from which models except that the table notes suggest all HRs are in fact SHRs. Are you, in fact, interested in a prognostic model, though, and not an aetiological one? This might be worth mentioning in the discussion.

I’m not sure that the interpretation on Lines 211–214 for the interaction with comorbidities is making it clear that those women with comorbidities would also have higher rates, I assume, of death from other causes, which for the analyses here would remove them from the risk set at the time of death (rather than censoring them with Cox’s PH model). This comes back to the aetiological versus prognostic modelling goal.

In terms of strengths, a large sample size (Line 232) isn’t a strength per se, but all other things being equal, it is more likely to produce sufficiently tight confidence intervals and (as you note, but without specifying the effect size of interest) provide more power to detect effects. Could you perhaps comment on the widths of the CIs here? Similarly, the length of follow-up (Lines 234–235) also isn’t a strength per se, but the number of events observed might be important when you consider model heuristics, such as those from Peduzzi, et al.

I appreciate you providing your list of potential confounders (Lines 235–237), which I hope will help other researchers when planning their studies. Your study limitations are very well described.

Additional comments

This is a well-performed and interesting study.

---

## Round 0.2 · Minor Revisions

Dear Authors,

Please address the remaining comments from the reviewers

Reviewer 1 ·

Basic reporting

I appreciate the efforts made by the authors. This new version is much better. Therefore, I have no further questions.

Experimental design

no comments

Validity of the findings

no comments

Additional comments

no comments

Reviewer 2 ·

Basic reporting

no comments

Experimental design

no comments

Validity of the findings

no comments

Additional comments

The authors has addressed all the queries I have raised on this manuscript.

·

Basic reporting

In the revised introduction where the authors nicely added the stroke part, it was neat to add a comparison between Chinese and Caucasian (line 74-75) by saying that "[t]he Chinese were reported to have higher stroke incidence than Caucasians." However, this sentence itself does not make a lot of sense without giving the incidence rate of stroke in Chinese or in Caucasians. Why use Caucasians as a reference point? Why not compare to other races or other countries? What's the exact number for the incidence rate? Say if the incidence rate in Caucasian is 1*E-100 % and that in Chinese is 1.1*E-100 %, I don't see why "there is an emerging need of stroke prevention for populations at high risk for stroke in Taiwan."

Experimental design

no comment

Validity of the findings

no comment

Additional comments

The authors did a very nice job addressing all the issues that were previously pointed out. It's just the added part in the introduction about stroke needs a bit polish to make a better sense.

·

Basic reporting

The manuscript is much improved and I have only a few relatively minor comments to make.

Rather than referring to “Cox proportional-hazard regression models with competing risks adjustment” (Lines 39–40), I think “Fine and Gray’s proportional subhazards model” or “competing risks survival analysis using the Fine and Gray model” or similar (this point is also clearer on Lines 158–159) would be more informative. There are several alternative competing risk models and Cox’s proportional hazards models can deal with competing risks through censoring, which could still be described as “Cox proportional-hazard regression models with competing risks adjustment” (and would be preferred over Fine and Gray if your questions were around aetiology rather than prognosis).

I appreciate your comments that you were focused on associations rather than causality (“However, our study primarily focused on the association of symptomatic menopausal transition with the development of subsequent stroke rather than a direct causal relationship between them.”), but I feel that this is inconsistent with the current version of your aim: “to determine whether symptomatic menopausal transition is a contributing factor to stroke.” (Lines 90–91), which I think is definite causal language. I think you need to revisit the wording for this aim.

I’m slightly confused by the sample size section (Lines 161–164), which doesn’t provide any information about the calculations, or make it clear that this was retrospective, and which is revisited in the discussion on Lines 272–275 without sufficient information to enable replication (what is the detectable effect size, what was the assumed proportion—to determine the standard error, or was the worst case of 0.5 used—and what was the level of significance used—presumably two-sided p<0.05, but this should still be made explicit?) I appreciate that this was retrospectively calculated (as I understand your response, data collection was completed at the time it was performed and so presumably no more data would have been collected irrespective of the calculation’s result), and in this case, it is the widths of the CIs that could be best used to evaluate whether or not the study was sufficiently powered. Related to this, while you state that “Additionally, the number of events observed may be important when modelling survival data (Austin et al., 2017; Peduzzi et al., 1995).” you do not explain how there is a strength to your study in this respect. I appreciate that the connection could be seen as simple enough for the reader to fill in the blanks here, but I think that you need to make your point here explicit (and I could be guessing incorrectly about what that exact point actually is).

You still refer to “control” on Lines 45, 47, 128, 150, etc. although you changed “control” to “comparison” on Line 32 and use “comparison” in Figure 2. Personally, I prefer “comparison group” to “control group”, but this should be consistent in any case.

Looking at Table 3, can you make it clear that the p-values for interactions refer to the second adjusted models and not the first (at least this is how I am reading the table).

Experimental design

No comment.

Validity of the findings

No comment.

---

## Round 0.3 · Minor Revisions

Dear Authors,Please proceed to do the minor revisions as per the second reviewer who has suggested minor revisions.Thanking you.

·

Basic reporting

no comment.

Experimental design

no comment.

Validity of the findings

no comment.

Additional comments

The authors nicely addressed the issue I pointed out.

·

Basic reporting

For the control/comparison group matter, the authors might have missed a few instances (which would be entirely understandable). See Line 291 (proof version) when you refer to misclassification as “controls” and the legend for Figure 1, column headings for Table 1, row label for Table 2 and the table notes also, column headings for Table 3, and the legend for Table 4.

On Line 44, I’d say “matched comparison participants” or “matched comparison women”, or “matched women without symptomatic menopausal transition” rather than “matched comparisons”. Also Lines 118, 173, and 225.

Line 73: I don’t think you need either of these commas (after “range” in each case).

Line 128: Perhaps “comparison participants”, “comparison women”, or “women without symptomatic menopausal transition” here rather than “comparisons”. Also Lines 153, 195, and 205.

Line 241: You can delete “an” here in “resulting in an weight gain”.

Experimental design

Retrospective power analyses using the observed effect size (as is the case here) turn out not to make any sense. If an effect of the observed size was statistically significant, then the actual study (or any other study with the same n) was powered at more than 50%; if an effect of the observed size was not statistically significant, then the study was powered at less than 50%. Looking at the sample size required still preserves this association for 50% power (if the actual sample size was larger than required for 50% power, the result was statistically significant and if the actual sample size was less than required for 50% power, the result was not statistically significant). This is slightly less obvious for 80% power, but there if the actual sample size was less than (approximately) half the calculated required sample size, the result was not statistically significant and if the actual sample size was more than (approximately) half the calculated required sample size, the result was statistically significant. In other words, no matter how we look at it, the retrospective power of a study using the observed effect size is determined simply by the p-value that was observed and so retrospective power using the observed effect size contributes absolutely no new information beyond that. If no other sample size calculation was performed, this fact should be stated instead of a retrospective calculation based on the observed effect and the point that the effect power of the study is communicated through the widths of confidence intervals made. This will also have implications for Line 277 (proof version).

Validity of the findings

No comment.

Additional comments

The authors have revised their manuscript well in response to my comments, and those of the other reviewers, with one important exception (under design) and a few minor details (under reporting). Well done!

---

## Round 0.4 · Minor Revisions

Please address these remaining comments

·

Basic reporting

No comment.

Experimental design

For the power calculation, as I noted previously "...retrospective power using the observed effect size contributes absolutely no new information beyond that. If no other sample size calculation was performed, this fact should be stated instead of a retrospective calculation based on the observed effect and the point that the effect[ive] power of the study is communicated through the widths of confidence intervals made." so I have to disagree that the retrospective power calculation (Lines 164-167) provides any information whatsoever. The sentence on Lines 278-280: "However, the retrospective power of this study calculated using an observed effect size was determined mainly by the P value which was already observed." would be correct for the analysis referred to in the power calculation if "mainly" was replaced with "entirely"!

As I suggested, if no sample size calculation was performed, this fact should simply be stated in the methods along with a comment in the discussion about whether the eventual precision (as measured via the 95% CI widths) was adequate or not.

Some references which explain why retrospective power should not be reported:

https://doi.org/10.1111/1467-9884.00139 "We suggest that it is nonsensical to make power calculations after a study has been conducted and a statistical decision has been made."

https://doi.org/10.1007/s10211-004-0095-z "As we have shown, retrospective power analyses have logical flaws and shortcomings when used for the interpretation of non-significant results." (It is rare to see retrospective power used when results are statistically significant and so adequate power is not in doubt.)

http://doi.org/10.2307/3802357 "Calculation of power after a completed study is appropriate for a future, related study, but because power is a probability, power cannot be applied to a completed study."

https://doi.org/10.1109/ICIC.2009.392 "We conclude that retrospective power calculation yield no additional insights if P value is available, and cannot be used to estimate the statistical power."

https://doi.org/10.1111/jch.13173 "Why it is nonsensical to use retrospective power analyses to conduct a postmortem on your study" (title)

https://doi.org/10.1016/j.ecns.2016.03.001 "There is a 1:1 correspondence of observed power and the p value calculated by hypothesis testing (Hoenig & Heisey,2001). This allows the researcher who does not find statistically significant results to conclude it was because her or his study was underpowered! Conversely, if an investigation results in significant findings, one can conclude the study was appropriately powered."

Validity of the findings

No comment.

Additional comments

Aside from the power calculation being retrospective, this is an excellent manuscript. Well done to the authors!

---

## Round 0.5 · accepted · Accept

Dear Authors, PeerJ is happy to accept the final revision of your manuscript. Congratulations!

·

Basic reporting

No comment

Experimental design

No comment

Validity of the findings

No comment

Additional comments

The revisions around the power of the study have addressed my comments from before and I have no new comments to make. Well done to the authors!